# Prognostic factors and prediction models for hospitalisation and all-cause mortality in adults presenting to primary care with a lower respiratory tract infection: a systematic review

Merijn H Rijk ,[1] Tamara N Platteel ,[1] Teun M C van den Berg,[1] Geert-Jan Geersing,[1] Paul Little ,[2] Frans H Rutten ,[1] Maarten van Smeden,[3] Roderick P Venekamp [1]

MvS and RPV contributed equally.

For numbered affiliations see end of article.

**Correspondence to**
Merijn H Rijk;
m.h.rijk@umcutrecht.nl

## ABSTRACT

**Objective** To identify and synthesise relevant existing prognostic factors (PF) and prediction models (PM) for hospitalisation and all-cause mortality within 90 days in primary care patients with acute lower respiratory tract infections (LRTI).

**Design** Systematic review.

**Methods** Systematic searches of MEDLINE, Embase and the Cochrane Library were performed. All PF and PM studies on the risk of hospitalisation or all-cause mortality within 90 days in adult primary care LRTI patients were included. The risk of bias was assessed using the Quality in Prognostic Studies tool and Prediction Model Risk Of Bias Assessment Tool tools for PF and PM studies, respectively. The results of included PF and PM studies were descriptively summarised.

**Results** Of 2799 unique records identified, 16 were included: 9 PF studies, 6 PM studies and 1 combination of both. The risk of bias was judged high for all studies, mainly due to limitations in the analysis domain. Based on reported multivariable associations in PF studies, increasing age, sex, current smoking, diabetes, a history of stroke, cancer or heart failure, previous hospitalisation, influenza vaccination (negative association), current use of systemic corticosteroids, recent antibiotic use, respiratory rate ≥25/min and diagnosis of pneumonia were identified as most promising candidate predictors. One newly developed PM was externally validated (c statistic 0.74, 95% CI 0.71 to 0.78) whereas the previously hospital-derived CRB-65 was externally validated in primary care in five studies (c statistic ranging from 0.72 (95% CI 0.63 to 0.81) to 0.79 (95% CI 0.65 to 0.92)). None of the PM studies reported measures of model calibration.

**Conclusions** Implementation of existing models for individualised risk prediction of 90-day hospitalisation or mortality in primary care LRTI patients in everyday practice is hampered by incomplete assessment of model performance. The identified candidate predictors provide useful information for clinicians and warrant consideration when developing or updating PMs using state-of-the-art development and validation techniques.

**PROSPERO registration number** CRD42022341233.

## STRENGTHS AND LIMITATIONS OF THIS STUDY

⇒ Both prognostic factors and prediction models were covered in this comprehensive systematic review.

⇒ The systematic search syntax included a validated prognostic search string to identify all relevant prognostic literature.

⇒ Prevailing guidelines were followed for data extraction (Checklist for Critical Appraisal and Data Extraction for Systematic Reviews of prognostic factors studies (CHARMS) and CHARMS), quality assessment (Quality in Prognostic Studies tool, Prediction Model Risk Of Bias Assessment Tool and Grading of Recommendations, Assessment, Development and Evaluations) and reporting (Preferred Reporting Items for Systematic Reviews and Meta-Analyses).

⇒ Substantial between-study heterogeneity hampered the pooling of the results in a meta-analysis.

## INTRODUCTION

Acute lower respiratory tract infections (LRTI) are common in primary care.[1 2] Uncomplicated LRTI episodes generally have a favourable natural course in otherwise healthy adults, and antibiotic treatment with amoxicillin confers only little benefit in terms of earlier symptom resolution, both overall and in higher-risk subgroups of patients.[3–5] The clinical spectrum of LRTI patients, however, is heterogeneous in terms of patient and disease-specific characteristics, and thus the risk of hospitalisation and death varies substantially. For example, observational evidence suggests that adverse outcomes such as hospitalisation and death occur in fewer than 1% of patients with uncomplicated LRTI,[5] whereas this risk is far more pronounced among those with community-acquired pneumonia (CAP).[6 7] However,

diagnosing CAP in a primary care setting is notoriously challenging,[8 9] leaving general practitioners (GP) with uncertainties about how to identify higher-risk groups among those presenting with LRTI symptoms.

Nevertheless, identification of LRTI patients at the highest risk of complications is pivotal as this might help GPs identify those in whom close follow-up or early (antibiotic) treatment is warranted. Currently, several scores exist to assist physicians in estimating the risk of adverse outcomes in LRTI patients, such as the Pneumonia Severity Index (PSI) and CURB-65 (a prediction rule including confusion, blood urea nitrogen, respiratory rate, blood pressure and age ≥65 years).[10 11] However, these prediction rules have been developed in secondary care and the inclusion of laboratory and/or radiographic features hampers applicability to outpatients. A modified version of the CURB-65 (CRB-65, not including blood urea nitrogen) has been proposed as primary care alternative, but its development in hospitalised patients raises uncertainty about its performance in outpatients.[10] An overview of existing primary care-specific prognostic factors (PF) and prediction models (PM) for adverse outcomes in LRTI patients could provide insight on (1) existing PFs or PMs suitable for use in clinical practice and (2) relevant PFs to include when developing a new or updating an existing model. We, therefore, aimed to synthesise existing knowledge on PFs and PMs for hospitalisation and mortality within 90 days in adult patients presenting to primary care with LRTI.

## METHODS

The protocol for this review was registered in PROSPERO and the review was reported according to the Preferred Reporting Items for Systematic Reviews and Meta-Analyses reporting guideline for systematic reviews.[12]

### Searches and study selection

The electronic databases of MEDLINE, Embase and the Cochrane Library were searched on 27 June 2022 for eligible studies. Keywords for 'respiratory tract infection', 'hospitalisation', 'death', 'adults' and 'primary care' were combined with a validated search string to identify relevant prognostic studies (online supplemental table S1).[13] Eligibility criteria were based on PICOTS (patients, index, comparison, outcome, timing, and setting) guidance; we included all studies on adults with LRTI (patients) in which PF or PM (index) for 90-day hospitalisation or all-cause mortality (outcome) were assessed on the day of diagnosis (timing) in primary care (setting).[14 15] Non-English articles or those without full-text availability were excluded.

After de-duplication, two review authors (MHR and TMCvdB) independently reviewed titles/abstracts of unique records retrieved from the electronic databases against the eligibility criteria. Next, the same authors independently reviewed the full text of potentially relevant articles for eligibility. Any disagreements were resolved

by discussion and, where necessary, by consulting a third reviewer (TNP). To increase the yield of potentially eligible articles, reference lists of the included articles were reviewed for eligibility. Studies assessing the prognostic value of PFs, developing a new PM, or externally validating a previously developed PM were eligible for inclusion.

### Data collection

Data from eligible PF and PM studies were extracted by MHR using standardised forms based on the Checklist for Critical Appraisal and Data Extraction for Systematic Reviews of PF studies (CHARMS-PF) and PM studies (CHARMS), respectively.[14 16] The main outcomes of interest were hospitalisation or death within 90 days after the initial LRTI consultation. For PF studies we extracted ORs, HRs or risk ratios with accompanying 95% CIs from both univariable and multivariable analyses. For those studies that did not report such estimates, these associations were calculated from crude data using Rothman's Episheet V.15 November 2021 wherever possible. For PM studies, model performance estimates in terms of discrimination (c statistic with accompanying 95% CI) and calibration (intercept and slope) were extracted. If calibration measures were absent, the total observed over expected (O/E) ratio with accompanying 95% CI of patients with the outcome was calculated wherever possible as a proxy of overall model calibration.[15] To this end, we used data on the distribution of subjects and outcomes across different PM risk strata and the corresponding predicted risks of the strata as reported in the original model development study. These calculations were performed in R V.4.2.2 using the 'metamisc' package.

Other data extraction items included study population, candidate predictors, sample size, (handling of) missing data, analysis (modelling method and/or model development and validation method) and interpretation of results. Where needed, corresponding authors of individual studies were contacted to request additional data.

### Assessment of risk of bias and applicability

Risk of bias assessment was performed by two authors (MHR and TNP) independently using the Quality in Prognostic Studies tool (QUIPS, assessing study participation, study attrition, PF measurement, outcome measurement and statistical analysis) and the Prediction Model Risk Of Bias Assessment Tool (PROBAST, covering the domains of participants, predictors, outcome and analysis) for PF and PM studies, respectively.[17 18] We excluded the 'study confounding' item from the QUIPS, since confounding is not applicable to prognostic research. The applicability of the included articles was assessed based on the PICOTS criteria.

### Synthesis methods

Due to substantial heterogeneity in domain, predictor and outcome definition, combining PF effect estimates or PM performance measures in meta-analyses was

considered inappropriate and results were therefore descriptively summarised, and stratified according to diagnosis (ie, LRTI vs pneumonia). Forest plots were created to visually summarise (1) univariable and multivariable effect estimates of PFs, and (2) discrimination and calibration performance measures of PMs. PFs were categorised into either demographics, patient history, healthcare use, medication use, signs and symptoms, laboratory tests or diagnosis. Promising candidate predictors were identified based on reported results of multivariable analysis in PF studies. For each promising PF and PM, we applied the Grading of Recommendations, Assessment, Development and Evaluations framework using prognostic research-specific guidance to rate the quality of evidence.[19 20]

## Patient and public involvement

There was no patient or public involvement in this study.

## RESULTS

### Study selection

The search strategy yielded a total of 3191 records. Removing duplicates left 2799 unique records (figure 1). After screening the title and abstract, 34 studies remained for full-text review. Of these, 19 were excluded (figure 1).

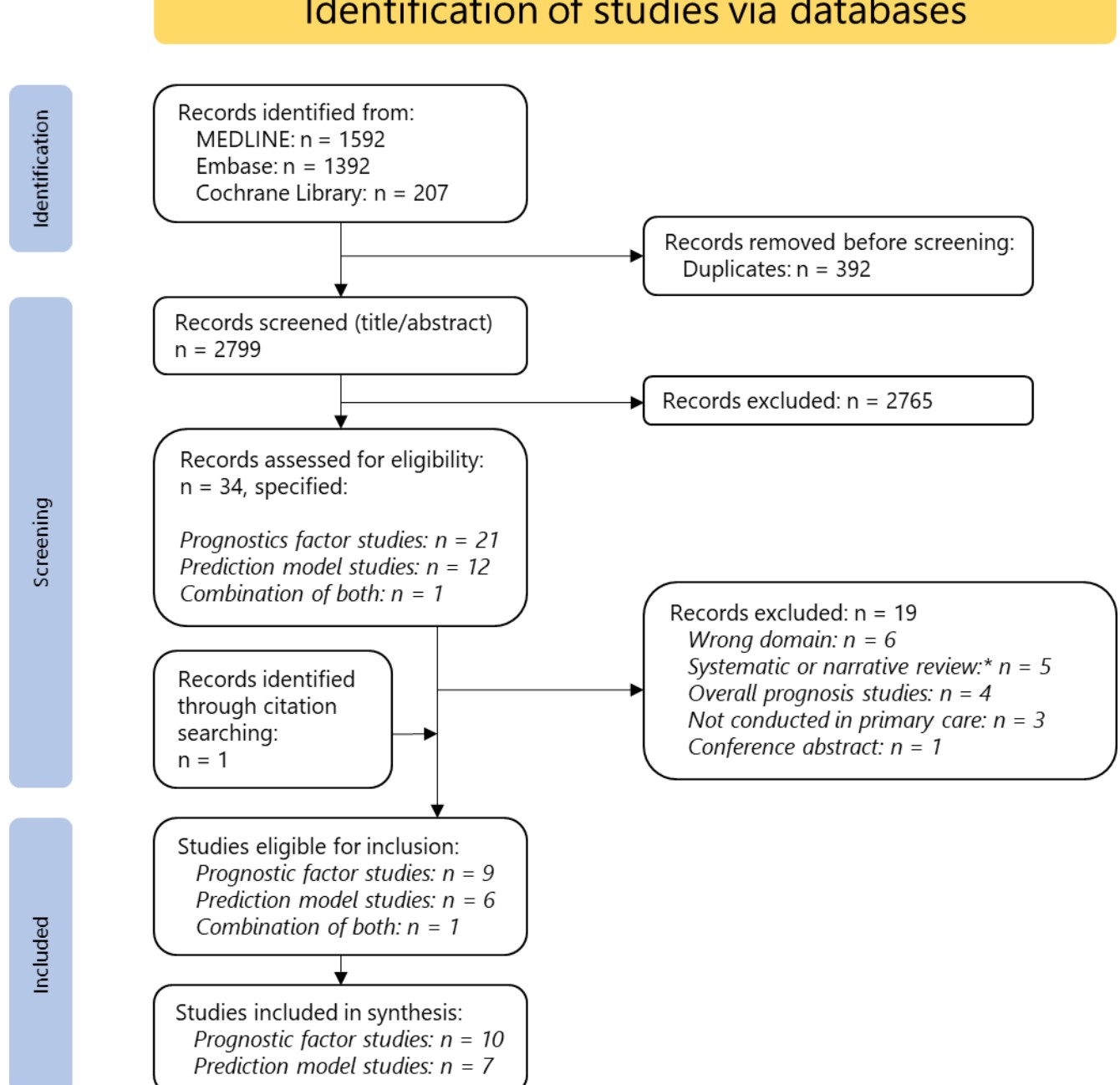

**Figure 1** Flow of articles through study. *Systematic reviews have been screened for eligible primary studies.

One additional publication was retrieved from screening reference lists,[21] leaving 16 publications for inclusion: 9 PF studies,[21–29] 6 PM studies[30–35] and 1 combination of PM and PF.[36]

## Study characteristics

The PF studies were conducted between 1997 and 2021 of which most (7/10) used a retrospective cohort design (table 1). The study population in terms of age (minimum age ranging from 1 to 80 years) and LRTI definition (ranging from suspected LRTI to diagnosis of CAP), as well as the hospitalisation rate (from 0.7% in LRTI patients to 76.5% patients with CAP) varied substantially across studies. Mortality was used as an outcome in four studies, hospitalisation in three studies and a composite of both in one study. Two studies also included deterioration of existing illness and late-onset pneumonia in addition to hospitalisation and mortality in their outcome,[23 36] while the timing of outcome assessment was unknown in one study.[29]

The PM studies were conducted between 2000 and 2013 and the majority (4/7) used a prospective cohort design. In total, nine different PMs could be included in the synthesis: CRB (CRB-65, not including age ≥65 years), CRB-65, CURB (CURB-65, not including age ≥65 years), CURB-65, PSI, a model developed by Bont *et al*, RISSC85 (a priori risk of poor outcome by country, inference in daily activities, number of years stopped smoking, severe sputum, presence of crackles, and diastolic blood pressure <85mmHg), CORB-75 (confusion, oxygen saturation ≤90%, respiratory rate ≥30/minute, systolic blood pressure ≤90mmHg or diastolic blood pressure ≤60mmHg, and age ≥75 years) and a model developed by Moore *et al* (table 2). Four studies developed a new PM[30 32 35 36] and five externally validated previously developed models,[31–35] of which the CRB-65 was validated in all five studies. Clinical heterogeneity was mostly attributed to age (minimum age ranging from 16 to 65 years) and LRTI definition (ranging from suspected LRTI to radiographically confirmed CAP). Hospitalisation rates varied between 0.5% in LRTI patients and 86.9% in patients with CAP. All PM studies used binary outcomes. Mortality was used as an outcome in three studies, hospitalisation in one study and a composite of both in one study. One study also included reconsultation with new or worsened symptoms in addition to hospitalisation in their outcome,[35] while another study included late-onset pneumonia in addition to hospitalisation and mortality.[36]

## Risk of bias assessment

Overall risk of bias was judged high for all studies (online supplemental figures S2 and S3). For PF studies, this was mostly due to issues in the statistical analysis domains. For instance, five studies selected predictors for multivariable analysis based on univariable associations,[21–25] and two included multiple episodes per patient but did not perform recurrent event analysis.[25 27] Furthermore, in some studies, an adequate definition of LRTI or PFs was lacking, and methods to account for missing data were mostly judged inadequate or not described.

Similarly, the risk of bias due to the issues in the analysis domain was judged high for all PM studies. In all model development studies, continuous predictors were dichotomised, and handling of missing data was judged inadequate or not reported. None of the PM studies reported any evaluation of model calibration.

## Prognostic factors

Three PF studies did not report estimates of univariable associations of PFs, and these could also not be calculated from the reported data.[24 27 28] The remaining PF studies reported univariable analysis of 69 PFs, of which sex, age and history of pulmonary disease were analysed most frequently (online supplemental figure S4). If reported in the primary study, absolute risks according to the presence and absence of individual PFs per study are presented in online supplemental table S5.

Estimates of multivariable associations were reported for 74 PFs (online supplemental figure S6), of which 21 were analysed in at least two studies. The number of covariates per multivariable analysis ranged from 3 to 29, and a common set of covariates across studies was lacking. Age was analysed in five studies, and all found increasing age to be associated with poor outcome.[23 24 27 28 36] Apart from increasing age, PFs that were identified as promising based on multiple multivariable analyses were sex (significant association in two/five studies), current smoking (three/four studies), diabetes (two/three studies), a history of stroke (two/three studies), cancer (three/three studies) or heart failure (two/two studies), hospitalisation in the previous year (two/two studies), current use of systemic corticosteroids (two/three studies), antibiotic use in the previous month (two/three studies), a respiratory rate ≥25/min (two/two studies) and diagnosis of pneumonia (two/two studies) (figure 2). Having received an influenza vaccination in the previous year was the only promising candidate predictor estimated to have a negative association, based on a significant multivariable association in two/three studies. The quality of evidence was judged low for age, hospitalisation in the previous year and diagnosis of pneumonia and very low for the remaining promising PFs (online supplemental table S7).

## Prediction models

Several PMs included predictors that were also identified as promising candidate predictors based on PF studies (online supplemental table S8). Sex, current smoking and influenza vaccination were the only promising candidate predictors that were not incorporated in any of the PMs. Predictors that were not identified as promising PFs but were incorporated in PMs include the history of lung, renal or liver disease, the extent to which issues interfere with daily activities, sputum or coryza, chest pain, confusion, presence of crackles, heart rate, blood pressure, oxygen saturation, body temperature, blood urea nitrogen and an overall GP severity assessment score.

**Table 1** Study characteristics of included prognostic factor studies

| Study | Source of data | Study population | Included episodes | Age, mean (SD) or median (IQR) | Outcome * | Hospitalisation, n (%) | Mortality, n (%) | Total outcome events, n (%) |
|---|---|---|---|---|---|---|---|---|
| Houston et al[21] | Community-based cohort, retrospective | Elderly (≥65 years) LRTI patients | 413 | NR | 30-day all-cause mortality | NR | 44 (10.7) | 44 (10.7) |
| Seppä et al[22] | Primary care cohort, prospective | Elderly (≥65 years) patients with LRTI suggestive of pneumonia | 950 | Survival: 76 (IQR 71–81), death: 80 (IQR 73–86) | LRTI-related mortality within 30 days | NR | 38 (4.0) | 38 (4.0) |
| Hak et al[23] | Primary care cohort, retrospective | Patients ≥60 years with an LRTI | 455 | 75 (SD 8.6) | Composite of: (1) dysregulation of diabetes, stroke, heart failure, MI, (2) hospitalisation, (3) all-cause mortality, all within 30 days | 21 (4.6) | 24 (5.3) | 65 (14.3) |
| Winchester et al[24] | Primary care cohort, retrospective | Patients ≥1 year with an LRTI | 151088 | 54 (IQR 37) | Respiratory infection-related admission or mortality within 3 months | 1147 (0.8) | 2126 (1.4) | NR |
| Van de Nadort et al[25] | Primary care cohort, retrospective | Elderly (≥80 years) LRTI patients | 860 | 85.2 (no SD) | Hospitalisation or death within 30 days | NR | 51 (5.9) | 109 (12.7) |
| Myles et al[26] | Primary care cohort, retrospective | Patients ≥40 years with an LRTI | 3681 | NR | All-cause mortality within 30 days | NR | 905 (24.6) | 905 (24.6) |
| Millett et al[27] | Primary care cohort, retrospective | Elderly (≥65 years) patients with CAP | 43576 | 81 (IQR 75–87) | Hospital admission within 28 days | 33321 (76.5) | NR | 33321 (76.5) |
| Moore et al[36] | Primary care cohort, prospective | Patients ≥16 years with acute cough attributed to LRTI | 28846 | NR | LRTI-related hospitalisation, mortality or late-onset pneumonia within 2–30 days | 196 (0.7) | 29 (0.1) | 325 (1.1) |
| Hamilton et al[28] | Primary care cohort, retrospective | Patients ≥50 years old with pneumonia and a history of lymphocyte test | 28556 | 76 (IQR 66–85) | 28-day mortality | NR | 2544 (8.9) | 2544 (8.9) |
| Martínez-Redondo et al[29] | Primary care cohort, prospective | Adults (≥18 years) with a COVID-19 pneumonia | 161 | 53.0 (no SD) | Hospital admission, no time window reported | 37 (23.0) | NR | 37 (23.0) |

*Time window of outcome is defined as days from first consultation within LRTI episode.
CAP, community-acquired pneumonia; COVID-19, coronavirus disease 2019; LRTI, lower respiratory tract infection; MI, myocardial infarction; NR, not reported.;

**Table 2** Study characteristics of included prediction model studies

| Study | Type of study | Model(s) | Source of data | Study population | Age, mean (SD) | Outcome* | Total outcome events, n (%) | Method of validation |
|---|---|---|---|---|---|---|---|---|
| Bont et al [30] | D+V | New | Primary care cohort, retrospective | Elderly (≥65) LRTI patients | 75.5 (no SD) | Hospitalisation or all-cause mortality within 30 days | D: 274 (8.7) V: 178 (7.2) | Internal (random split-sample) and external |
| Bont et al [31] | V | CRB-65 | Primary care cohort, prospective | Elderly (≥65) patients with CAP | 77.3 (no SD) | Mortality within 30 days | 11 (3.5) | External |
| Ochoa-Gondar et al [32] | V | PSI, CURB-65, CRB-65 | Population-based cohort, retrospective | Elderly (≥65) patients with radiographically confirmed CAP | 77.4 (7.6) | Mortality within 30 days | 80 (13.6) | External |
| Francis et al [33] | V | CRB-65 | Primary care cohort, prospective | Adults (≥18) presenting with acute cough or suspected LRTI | 49.3 (16.5) | Hospitalisation within 28 days | 10 (2.9) | External |
| Ochoa-Gondar et al 2013 [34] | D+V | CORB-75 (new), CRB-65 | Population-based cohort, retrospective | Elderly (≥65) patients with radiographically confirmed CAP | 78.1 (7.7) | Mortality within 30 days | D: 46 (13.1) V: 34 (13.1) | Internal-external (temporal, CORB-75), external (CRB-65) |
| Bruyndonckx et al [35] | D (RISSC85)+V (all models) | RISSC85 (new), PSI (stage I), CRB, CRB-65, CURB, CURB-65 | Primary care cohort, prospective | Adults (≥18) with symptoms suggestive of LRTI | 50 (17) | Reconsultation with new/worsened symptoms or hospitalisation within 28 days | 521 (20.0) | Internal (random split sample, RISSC85), external (other models) |
| Moore et al [36] | D+V | New | Primary care cohort, prospective | Patients ≥16 years with acute cough attributed to LRTI | NR | Hospitalisation, mortality, or clinical diagnosis of pneumonia within 2–30 days | 325 (1.1) | Internal (bootstrapping) |

*Time window of outcome is defined as days from first consultation.
CAP, community acquired pneumonia; CORB-75, confusion, oxygen saturation ≤90%, respiratory rate ≥30/minute, and systolic blood pressure ≤90mmHg or diastolic blood pressure ≤60mmHg, and age ≥75 years; CRB-65, Confusion, respiratory rate ≥30/minute, systolic blood pressure ≤90mmHg or diastolic blood pressure ≤60mmHg, and age ≥65 years ; CRB, Confusion, respiratory rate ≥30/minute, and systolic blood pressure ≤90mmHg ; CURB-65, Confusion, blood urea nitrogen >7 millimole/liter, respiratory rate ≥30/minute, systolic blood pressure ≤90mmHg or diastolic blood pressure ≤60mmHg, and age ≥65 years ; CURB, Confusion, blood urea nitrogen >7 millimole/liter, respiratory rate ≥30/minute, and systolic blood pressure ≤90mmHg or diastolic blood pressure ≤60mmHg; D, development; LRTI, lower respiratory tract infection; NR, not reported; PSI, Pneumonia Severity Index; RISCC85, a priori risk of poor outcome by country, inference in daily activities, number of years stopped smoking, severe sputum, presence of crackles, and diastolic blood pressure <85mmHg; V, validation.

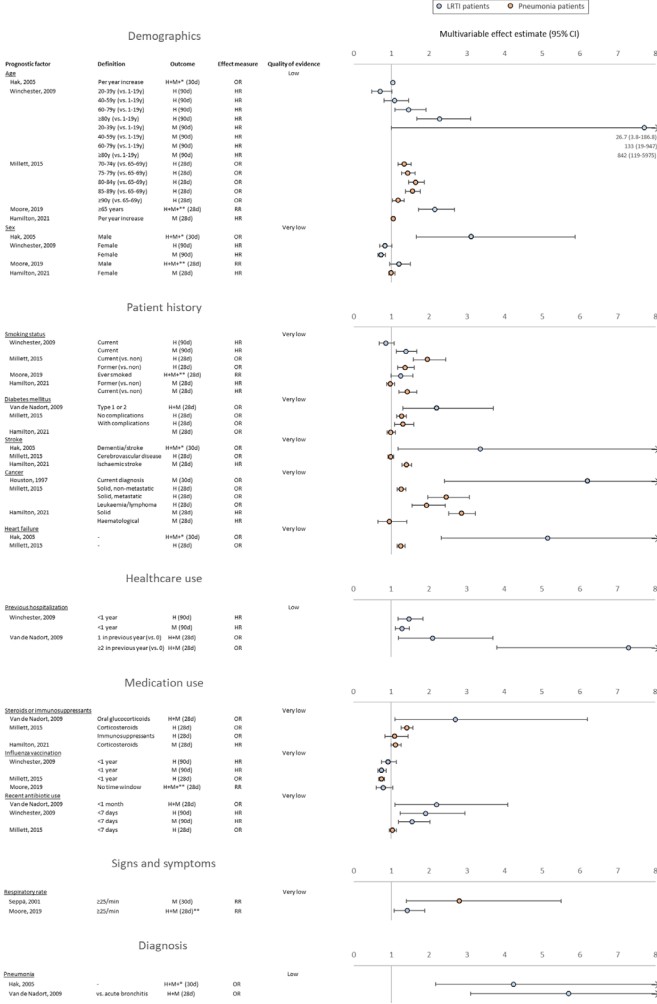

**Figure 2** Overview of most promising candidate predictors for hospitalisation or mortality within 90 days based on multivariable analysis. Blue coloured points: studies on LRTI patients. Orange coloured points: studies on patients with pneumonia. Quality of evidence: as judged based on the Grading of Recommendations, Assessment, Development and Evaluations framework. *Composite outcome also includes dysregulation of diabetes, stroke, heart failure, myocardial infarction. **Composite outcome also includes late-onset pneumonia. CAP, community-acquired pneumonia; H, hospitalisation; LRTI, lower respiratory tract infection; M, mortality; min, minute; RR, risk ratio.

Four studies developed a new PM (three for use in LRTI and one for patients with CAP), in which the number of events per predictor ranged from 3.3 to 13.7.[30 34–36] C statistics ranged from 0.63 (95% CI 0.61 to 0.67) to 0.82 (95% CI 0.75 to 0.88) after (internal or external) model validation (figure 3). One study performed external validation,[30] whereas other studies only conducted internal(-external) validation.[34–36] None reported measures of model calibration, while a total O/E ratio could not be calculated for any model.

All five PM validation studies externally validated the CRB-65.[31–35] C statistics for 30-day mortality in patients with CAP (range outcome events: 11–80) ranged from 0.72 (95% CI 0.66 to 0.78) to 0.79 (95% CI 0.65 to

0.92)[31 32 34] (figure 3). The c statistic for 28-day reconsultation or hospitalisation in LRTI patients was substantially lower: 0.53 (95% CI 0.51 to 0.56).[35] None reported measures of model calibration. A total O/E ratio could be calculated for the CRB-65 in four studies, ranging from 0.67 (95% CI 0.26 to 1.08) to 1.42 (95% CI 1.13 to 1.71).

Other validated PMs were PSI, CURB-65, CURB and CRB. C statistics for 28-day reconsultation or hospitalisation in LRTI patients ranged from 0.51 (95% CI 0.50 to 0.54) to 0.53 (95% CI 0.50 to 0.55).[35] Calibration measures were not reported and total O/E ratios could not be calculated. C statistics of PSI and CURB-65 for 30-day mortality in patients with CAP were 0.73 (95% CI 0.67 to 0.79) and 0.67 (95% CI 0.61 to 0.74), respectively.[32] Model calibration measures were not reported, while the total O/E ratios were estimated at 1.46 (95% CI 1.16 to 1.76) and 1.92 (95% CI 1.53 to 2.32), respectively. The quality of evidence was judged low for CRB-65 and very low for all other PMs included in this systematic review (online supplemental table S9).

## DISCUSSION
### Summary of main findings
This systematic review summarised evidence on PFs and PMs for hospitalisation and mortality within 90 days in patients presenting to primary care with LRTI. The risk of bias in included studies was judged high, mainly due to issues in the analysis domains. Of all PFs analysed, 13 were identified as most promising; increasing age, sex, current smoking, diabetes, a history of stroke, cancer or heart failure, hospitalisation in the previous year, current use of systemic corticosteroids, antibiotic use in the previous month, a respiratory rate ≥25/min and diagnosis of pneumonia were associated with poor outcome whereas seasonal influenza vaccination was estimated to have a negative association. Most of these promising candidate predictors have also been incorporated into the identified PMs. The secondary care-derived CRB-65, predicting mortality in patients with CAP and a newly developed PM by Bont *et al*[30] predicting hospitalisation and mortality in elderly LRTI patients both showed promising results after external validation. However, none assessed model calibration, leaving the precision of predicted risks unknown.

### Comparison with existing literature
During the coronavirus disease 2019 (COVID-19) pandemic, many studies have focused on PFs associated with poor outcomes in patients with COVID-19, and some were similar to those identified in our review of all-cause LRTI patients. A recent, field-wide systematic review and meta-analysis of PFs for adverse outcomes in patients with COVID-19 found cancer, insulin use and smoking—among other factors—to be associated with mortality and age with both hospitalisation and mortality.[37] Another community-based systematic review of patients with European COVID-19 found male sex, heart failure and diabetes to be associated with hospitalisation, whereas stroke, heart failure and neurological disease

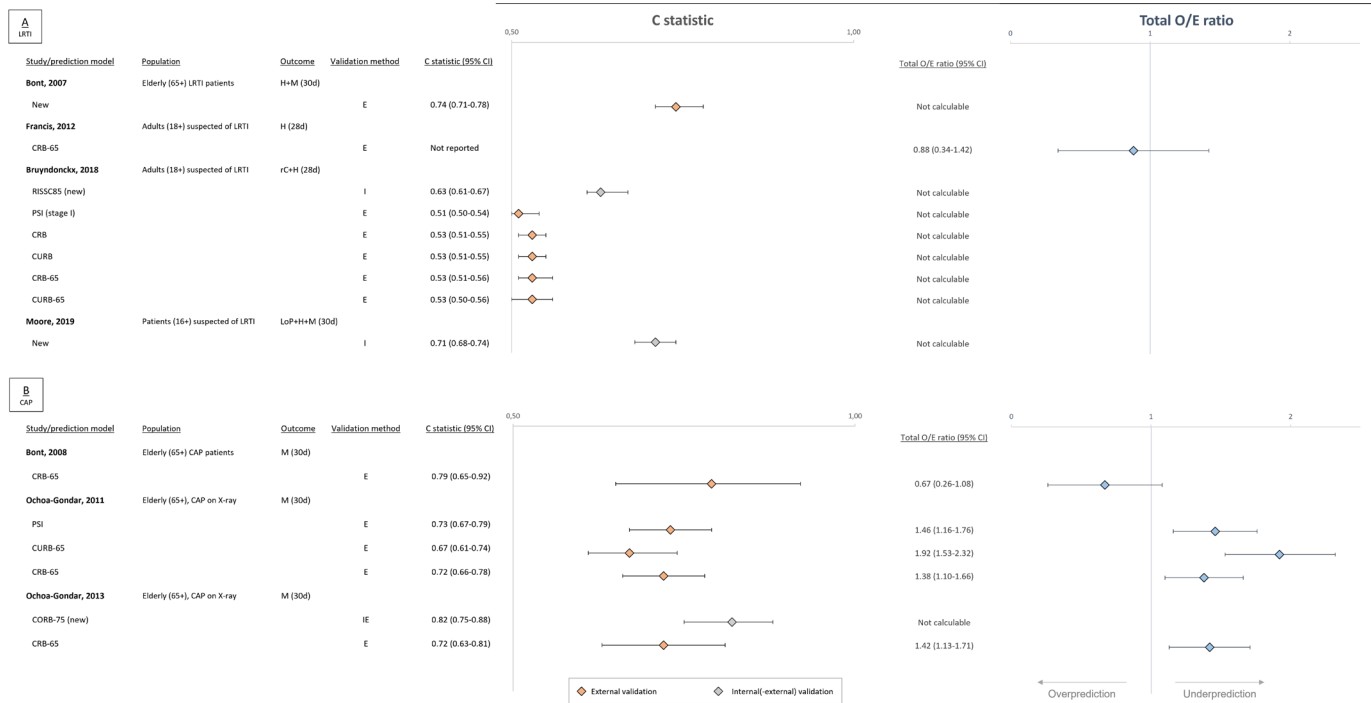

**Figure 3** Overview of discrimination and calibration performance measures of the included prediction models. (A) Performance of prediction models evaluated in LRTI patients. (B) Performance of prediction models evaluated in community-acquired pneumonia patients. CAP, community-acquired pneumonia; CORB-75, confusion, oxygen saturation ≤90%, respiratory rate ≥30/minute, and systolic blood pressure ≤90mmHg or diastolic blood pressure ≤60mmHg, and age ≥75 years; CRB(-65), confusion, respiratory rate ≥30/minute, systolic blood pressure ≤90mmHg or diastolic blood pressure ≤60mmHg, and age ≥65 years (and age ≥65 years); CURB(-65), confusion, blood urea nitrogen >7 millimole/liter, respiratory rate ≥30/minute, systolic blood pressure ≤90mmHg or diastolic blood pressure ≤60mmHg (and age ≥65 years); E, external; H, hospitalisation; I, internal; IE, internal external; LoP, late-onset pneumonia; LRTI, lower respiratory tract infection; M, mortality; O/E, observed over expected; PSI, Pneumonia Severity Index; rC, reconsultation; RISSC85, a priori risk of poor outcome by country, inference in daily activities, number of years stopped smoking, severe sputum, presence of crackles, and diastolic blood pressure <85mmHg.

were associated with increased mortality risk.[38] Although we did not identify studies that assessed the prognostic value of specific viral respiratory pathogens such as SARS-CoV-2, respiratory syncytial virus and influenza, it is likely that the specific pathogen is a relevant PF and point-of-care tests for respiratory pathogens are increasingly available in the post-pandemic era. The potential usefulness of such point-of-care tests is supported by evidence of patients with a viral LRTI admitted to the emergency room in whom specific viral pathogens were associated with an increased risk of hospitalisation.[39]

Some of the identified PMs have been incorporated in LRTI guidelines—mainly those focusing on CAP—as additional tools to aid patient management decisions. The CAP guideline issued by the American Thoracic Society (ATS) and the Infectious Diseases Society of America (IDSA) supports the use of PSI or CURB-65—in addition to clinical judgement—to determine the appropriate level of care, that is, inpatient versus outpatient management.[40 41] Although this guideline applies to all levels of healthcare, the value of the suggested models for GPs is limited since the models' laboratory and radiographic features are not routinely available in primary care. The UK National Institute for Health and Care Excellence (NICE) guideline on pneumonia advocates the use of

CRB-65; it recommends that GPs consider hospital admission in patients with a score of ≥1.[42] Guidelines vary with regard to the incorporation of individual factors that are associated with poor outcome. The ATS/IDSA guideline and the Dutch primary care guideline both list tachypnoea and confusion as indicators of disease severity, while the Dutch guideline also emphasised the importance of PFs such as age, history of heart failure and diabetes.[40 43] The NICE guideline does not explicitly highlight individual factors associated with poor outcome in adult patients.[42]

Similar to our findings, a review of the performance of CRB-65 for predicting 30-day mortality among patients with CAP in community and hospital settings concluded that this model has been insufficiently validated for use in primary care.[44] Although the population of this review slightly differed from ours by also including patients who visited the emergency department, a similarly low number of outcome events was found. Contrary to the slight overprediction reported in this review, our synthesis of more recent studies suggested underprediction of risks based on total O/E ratios. However, in the absence of a proper assessment of model calibration, the true precision of predicted risks remains to be elucidated.

## Strengths and limitations

By covering both PFs and PMs in a single review, we have provided a comprehensive overview of available prognostic literature on primary care LRTI patients. We prospectively registered our review protocol, conducted a broad search with the incorporation of a validated prognostic search string and thoroughly reviewed citations of eligible studies to assure all relevant literature was included in the synthesis. By including various prognostic study designs, our synthesis provides a comprehensive overview of all prognostic evidence on primary care LRTI patients. Studies assessing the prognostic value of individual factors, and developing a new or validating an existing model all provide insight on important PFs that could be used in future prognostic research, whereas studies validating a new or previously developed model such as CRB-65 provide insight on the suitability for its use in clinical practice.

Some limitations should, however, be acknowledged. First, our findings should be interpreted with caution since the risk of bias was judged high for all included studies, and the quality of the evidence was judged low to very low for the promising PFs and PMs. Second, due to substantial heterogeneity in domain, predictor and outcome definitions between studies, we refrained from conducting meta-analyses of PF effect estimates or PM performance measures, which makes direct comparison of study results difficult. It is plausible that PF effect estimates for poor outcome vary according to age, diagnosis and disease severity (a priori risk of event). For example, half of the participants in Hak et al[23] had either pneumonia or an exacerbation of chronic obstructive pulmonary disease, whereas Moore et al[36] had fewer than 2% of such patients. Furthermore, one study included patients aged >1 year (median age of included participants: 54 years (IQR: 37)) but did not specifically report separate results for age strata.[24] Given the median age, we considered including this study on PF in our review justified. Consistency in effect estimates would have further facilitated direct comparison of PFs, but converting the various reported estimates to ORs appeared not possible. Although our synthesis of PFs provides some useful information for clinicians, it merely informs future research aimed at developing or updating PMs. Third, we excluded most studies on patients with a specific respiratory pathogen (eg, SARS-CoV-2 or influenza) since these included patients with both upper and lower respiratory tract infections and did not report stratified results for LRTI. As such, we were unable to present any respiratory pathogen-specific results regarding PF for poor outcome in patients presenting to primary care with LRTI. However, since GPs are typically unaware of the specific pathogen in LRTI patients, such results would have limited applicability to current clinical practice. Lastly, PM building techniques were generally suboptimal which increased risk of bias, and none of the identified PM studies assessed model calibration. In contrast to discrimination, calibration generally receives little attention when assessing PM performance, whereas well-discriminating but poorly calibrated models can lead to imprecise, misleading predictions.[45] Total O/E ratios calculations provided some insight into mean model calibration—the most basic level in the hierarchy of calibration assessment—but this is insufficient to provide guidance on the safe use of a PM in clinical practice.[46]

## Implications for research and clinical practice

Based on our synthesis, implementation of existing PMs for individualised risk prediction of 90-day hospitalisation or all-cause mortality in primary care LRTI patients in everyday practice is hampered by incomplete assessment of model performance. The identified candidate predictors provide useful information for clinicians about factors associated with poor outcome in primary care LRTI patients and warrant consideration when developing or updating PMs using state-of-the-art development and appropriate validation techniques.

**Author affiliations**
[1]Department of General Practice, Julius Centre for Health Sciences and Primary Care, University Medical Centre Utrecht, Utrecht, The Netherlands
[2]Primary Care and Population Science, University of Southampton, Southampton, UK
[3]Julius Center for Health Sciences and Primary Care, University Medical Center Utrecht, Utrecht, The Netherlands

**Contributors** MHR, TNP, MvS and RV were involved in the conceptualisation and design of the study. MHR, TMCvdB and TNP conducted the study selection procedure. Data of eligible studies was extracted by MHR. MHR, TNP and MvS performed the risk of bias assessment. MHR wrote the first draft of the manuscript, which was initially reviewed by TNP, MvS and RV. MHR, TNP, TMCvdB, G-JG, PL, FHR, MvS and RV reviewed and approved the final manuscript. As guarantor, MHR declares that this manuscript is an honest, accurate, and transparent account of the study, and accepts full responsibility for its conduct and reporting.

**Funding** This study was funded by ZonMw (grant number 08391052110003). The funder played no role in study design, data collection, analysis and interpretation of data, or the writing of this manuscript.

**Competing interests** None declared.

**Patient and public involvement** Patients and/or the public were not involved in the design, or conduct, or reporting, or dissemination plans of this research.

**Patient consent for publication** Not applicable.

**Ethics approval** Not applicable.

**Provenance and peer review** Not commissioned; externally peer reviewed.

**Data availability statement** Data are available upon reasonable request. Study materials (eg, template data collection forms, data extracted from included studies) are available from the corresponding author upon reasonable request.

**ORCID iDs**
Merijn H Rijk http://orcid.org/0000-0003-4190-2126
Tamara N Platteel http://orcid.org/0000-0001-5122-0079
Paul Little http://orcid.org/0000-0003-3664-1873

Frans H Rutten http://orcid.org/0000-0002-5052-7332
Roderick P Venekamp http://orcid.org/0000-0002-1446-9614

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
