## [Reviewer comments · BMJ Open]

ARTICLE DETAILS

TITLE (PROVISIONAL)	Prognostic factors and prediction models for hospitalisation and all-cause mortality in adults presenting to primary care with a lower respiratory tract infection: a systematic review
AUTHORS	Rijk, Merijn; Platteel, Tamara; van den Berg, Teun; Geersing, Geert-Jan; Little, Paul; Rutten, Frans; van Smeden, Maarten; Venekamp, Roderick

VERSION 1 – REVIEW

REVIEWER	Davido, Benjamin APHP, Maladies Infectieuses, Hôpital Universitaire Raymond-Poincaré
REVIEW RETURNED	20-Jul-2023

GENERAL COMMENTS	Authors present a summarize of all scores (models) and predictors of LRTI before the Covid pandemic in an ambulatory care settings. Authors conclude that the candidate predictors provide useful information for primary care and might help to develop new models, albeit being hampered by incomplete assessment of model performance. Some questions / concerns / suggestions: 1) Page 6 line 10 "antibiotics confer ...", actually it should be rephrased by amoxicillin confer . Authors cannot extrapolate the cited reference to other molecules such as macrolides for instance2) Page 6, line 16 I would suggest swapping the following paragraph for clarification : "Within the clinical spectrum...." and "However in patients". It makes sense since you are referring of CAP.3) Page 10 :study characteristics. My main concerns is about mixing pediatrics children and adults, especially for the prognosis analysis. Some will say that the analysis of age and comorbidities is then uninterpretable.4) Page 12 Line 41 (Four studies developed a new PM). What is then this "new" factor being studied? can you explain ? is it reliable ?5) Page 14 , line 15: once again "AGE". Which age ? more than 65 ?6) Page 14 line 17: "current use of steroids". Considering new studies (including one JAMA) recently showed interest about steroids in severe pneumonia, are you referring to long-term of steroids ? moreover inhaled steroids ?7) Page 14 line 22 "influenza vaccination". What about pneumococcal vaccine ? do we have any data in the literature ?8) About the paragraph entitled "Comparison with existing literature", you develop and entire comparison about Covid, similarly some studies focused on factors associated with viral
--

	LRTI including influenza and other viruses. I suggest you read the Superfluous study designed at the ER. 9) Page 15 about the whole guidelines section ; don't you think that in addition to factors or models, use of point of care in the post Covid era could be relevant to assess the potential severity of infection accord to virus (in comparison to bacteria ?) 10) Page 16: Line 38 "Third.we excluded most studies regarding SARS-Cov2 and influenza.". As such you might have included studies dealing with other viruses such as RSV, metaPneumovirus or PIV, not only bacterial infections. Is it really a limitation?
--	---

REVIEWER	Filipow, Nicole University College London Institute of Child Health, Physiotherapy, RCCA, III
REVIEW RETURNED	18-Aug-2023

GENERAL COMMENTS	The systematic review by Rijk et al. covers studies of prognostic factors and prediction models in lower respiratory tract infections in primary care. It summaries the studies descriptively by reporting on risk of bias, significant predictors, and performance of predictive models. The authors conclude that the papers reviewed are largely unreliable, but have some common important predictors amongst them. Thank you to the editor for the opportunity to review this paper. It is well written and easy to follow, and is an important review that synthesizes the prognostic information in LTRIs. However, I have some questions surrounding the study design in relation to the aims that should be clarified in a revision. Major Comments 1. The authors state that the aim is to “synthesize existing PFs and PMs for hospitalisation and mortality within 90 days in adult patients presenting to primary care with LRTI”, but it is not clear for what purpose. The intro makes the point that it is important to predict whether a LTRI is viral or CAP, but the results include studies that only assess risk factors in CAP patients that have already been diagnosed. The intro also says that current indices (PSI & CURB-65) are no good for primary care, but the results include studies that investigated CRB-65 (It is not clarified if this predictive model also uses radiological/lab measures). A clearer aim on why the specific papers were reviewed should be considered. 2. In line with the above it is not clear what the aim of reviewing both new PMs and the evaluation of standard PMs was. Page 11, line 44 says 9 different models were identified – but it appears that only actually 4 were identified, in addition to the CRB-65 that was evaluated in 5 different papers. If the aim was to synthesise the prognostic factors, the different models would help with that to understand what factors are included in and are important in each model. Conversely, the 5 studies evaluating the CRB-65 do not offer novel predictive information, but rather the synthesis of these studies would infer prognostic performance of that model and provide evidence of whether it should be used in primary care. Some clarification of the aims and purpose of the study within the intro would help address these points. Minor Comments
--

	1. CRB-65 should be defined in the abstract as a previously defined/standard model. 2. P11 Line 54- Reported hospitalisation rates between 0.5 and 86.9% appears a big range for primary care. Should be clarified that this includes CAP only studies or separate CAP and LTRI from each other. 3. P11 Line 55-60 – The outcomes listed for each study does not add up to the overall number of PM studies (9) 4. P13 first paragraph - might be useful to show how many studies found the significant association for each predictor compared to how many studies investigated that predictor. For example: “multiple multivariable analyses were sex (five / five studies), current smoking (four /three studies, etc.) 5. Figure 2 is difficult to read. Perhaps placing the CAP and LTRI models side by side might be clearer, or summarising the predictor variables into a single row each, showing different colour points for different studies/categories.
--	---

REVIEWER	Tortosa , Fernando Pan American Health Organization, EIH
REVIEW RETURNED	11-Nov-2023

GENERAL COMMENTS	I have thoroughly reviewed your manuscript concerning prognostic factors for hospitalization and mortality in individuals with lower respiratory tract infections. I commend your efforts in addressing this critical topic. However, I would like to provide some feedback and suggestions to enhance the comprehensiveness and reliability of your findings. Literature Search and Inclusion of Non-English Publications and Grey Literature: It is crucial to understand whether the literature search conducted for this study was exhaustive. Did it include publications in languages other than English? Including non-English studies can provide a more comprehensive view, especially considering the global impact of lower respiratory tract infections. Additionally, what grey literature was included in the study? This is particularly important in this field where health disparities, including those related to gender, age, socioeconomic status, living conditions, and prevalence of smoking, play a significant role. A broader scope in literature inclusion ensures that these disparities are well represented. Adjustment of Prognostic Factors Through Multivariate Analysis: It’s essential to emphasize whether these prognostic factors have been adjusted via multivariate analyses in the various studies reviewed. This adjustment is crucial to isolate the effect of each factor and understand its independent contribution to the risk of hospitalization and mortality. Baseline Risk Assessment: The manuscript should consider the baseline risk of hospitalization and death in patients with these infections. How is this baseline risk difference valued in your presentation? Presenting both relative and absolute estimates of the number of events with and without the risk factor, modeling different baseline risks (e.g., risk of death in children vs. elderly), can provide a clearer understanding of the real impact of these factors. Application of GRADE Methodology:
---

	The application of the GRADE methodology for each prognostic factor and outcome is recommended. This approach will help define the level of certainty we have for each effect estimate. This methodology is crucial for advancing towards prognostic scores and being able to estimate adjusted values and confidence in the effects for each estimation. Your research contributes significantly to understanding the dynamics of lower respiratory tract infections. These suggestions are intended to strengthen the robustness of your conclusions and enhance the paper's utility for clinicians and policymakers.
--	---

VERSION 1 – AUTHOR RESPONSE

Reviewer #1:

Authors present a summarize of all scores (models) and predictors of LRTI before the Covid pandemic in an ambulatory care settings. Authors conclude that the candidate predictors provide useful information for primary care and might help to develop new models, albeit being hampered by incomplete assessment of model performance.

Some questions / concerns / suggestions:

Comment 1. "Page 6 line 10 "antibiotics confer ...", actually it should be rephrased by amoxicillin confer. Authors cannot extrapolate the cited reference to other molecules such as macrolides for instance."

Response: We agree with the reviewer that the cited reference only provides evidence on the benefits of amoxicillin treatment in primary care patients with lower respiratory tract infections (LRTI), and that this cannot necessarily be extrapolated to other classes of antibiotics.

Changes: As suggested, we have changed the sentence in the Introduction section to:

'Uncomplicated LRTI episodes generally have a favourable natural course in otherwise healthy adults, and antibiotic treatment with amoxicillin confers only little benefit in terms of earlier symptom resolution, both overall and in higher-risk subgroups of patients.'

Comment 2. "Page 6, line 16 I would suggest swapping the following paragraph for clarification: "Within the clinical spectrum...." and "However in patients". It makes sense since you are referring of CAP."

Response: We thank the reviewer for this valuable suggestion. Prompted by this comment and by comment #1 of reviewer #2, we have now substantially revised the Introduction section.

Comment 3. "Page 10: study characteristics. My main concern is about mixing paediatrics children and adults, especially for the prognosis analysis. Some will say that the analysis of age and comorbidities is then uninterpretable."

Response: We understand this reviewers' concern and acknowledge that mixing paediatric and adult populations may lead to difficulties in interpreting the results. However, in our synthesis we only included one study of both paediatric and adult patients, i.e. aged 1 year and older. Albeit this study did not specifically report results for the subgroup of adult patients, the study findings are largely applicable to the adult population since the median age of the study population was 54 years (interquartile range: 37).

Since the objective of our systematic review was to synthesize the wide body of prognostic evidence of adult primary care LRTI patients and we did not perform a meta-analysis of effect estimates, we believe that inclusion of this specific study of prognostic factors is justified and does not lead to problems with interpretation of our main findings.

Changes: We have now explicitly elaborated upon the inclusion of this specific study in the Discussion section: 'Furthermore, one study included patients aged >1 year (median age of included participants:

54 years [IQR 37]) but did not specifically report separate results for age strata. Given the median age, we considered including this study on prognostic factors in our review justified.'

Comment 4. "Page 12 Line 41 (Four studies developed a new PM). What is then this "new" factor being studied? Can you explain? Is it reliable?"

Response: Four of the included prediction model studies developed a new prediction model (PM). These include RISSC85, CORB-75, a model developed by Bont et al. and a model developed by Moore et al. These four are indicated as 'new' models in Figure 3. The performance of these new models in term of discrimination (c statistic) and overall calibration (observed over expected ratio) is summarised in Figure 3.

Comment 5. "Page 14, line 15: once again "AGE". Which age? more than 65?"

Response: The studies included in our synthesis of prognostic factors did not use a common definition of age; some studies modelled age as a continuous variable whereas others applied various cut-off values. Nevertheless, all studies found an association between increasing age and poor outcome in primary care LRTI patients.

Changes: To clarify that increasing age has a positive association with poor outcome in the included studies, we rephrased 'age' as promising prognostic factor to 'increasing age' throughout the manuscript.

Comment 6. "Page 14 line 17: "current use of steroids". Considering new studies (including one published in JAMA) recently showed interest about steroids in severe pneumonia, are you referring to long-term of steroids? Moreover inhaled steroids?"

Response and changes: We agree with the reviewer that it was unclear whether 'current use of steroids' referred to inhaled and/or systemic corticosteroids. Based on the included studies, current use of systemic corticosteroids at diagnosis of LRTI was identified as promising prognostic factor for poor outcome.

Three studies assessed the prognostic value of inhalation corticosteroid use on poor outcome in primary care LRTI patients (Figure S4 and S5, supplementary data). Since only one of these studies reported a significant (negative) association with poor outcome based on multivariable analysis, we did not include inhalation corticosteroid use as promising prognostic factor. We have therefore changed 'current use of steroids' to 'current use of systemic corticosteroids' throughout the manuscript.

Comment 7. "Page 14 line 22 "influenza vaccination". What about pneumococcal vaccine? Do we have any data in the literature?"

Response: As shown in Figure S4 and S5 (supplementary data), three prognostic factor studies assessed the association between pneumococcal vaccination and poor outcome in LRTI patients. None of these studies reported a significant association based on multivariable analysis, and we therefore did not include pneumococcal vaccination as promising prognostic factor.

Comment 8. "About the paragraph entitled "Comparison with existing literature", you develop an entire comparison about Covid, similarly some studies focused on factors associated with viral LRTI including influenza and other viruses. I suggest you read the Superfluous study designed at the ER."

Comment 9. "Page 15 about the whole guidelines section: don't you think that in addition to factors or models, use of point of care in the post Covid era could be relevant to assess the potential severity of infection accord to virus (in comparison to bacteria?)"

Response and changes: We thank the reviewer for these comments and have added the Superfluous study – on viral LRTI patients admitted to the emergency room – and the potential of point-of-care testing for respiratory pathogens to our (comparison with existing literature paragraph of) the Discussion section:

'Although we did not identify studies that assessed the prognostic value of specific viral respiratory pathogens such as SARS-CoV-2, respiratory syncytial virus and influenza, it is likely that the specific pathogen is a prognostic factor, and point-of-care tests for respiratory pathogens are increasingly available in the post-pandemic era. The potential usefulness of such point-of-care tests is supported by evidence of patients with a viral LRTI admitted to the emergency room in whom specific viral pathogens were associated with increased risk of hospitalisation.'

Comment 10. "Page 16: Line 38 "Third, we excluded most studies regarding SARS-Cov-2 and influenza.". As such you might have included studies dealing with other viruses such as RSV, metapneumovirus or PIV, not only bacterial infections. Is it really a limitation?"

Response: By addressing this limitation, we intended to state that we could not include any studies on patients with a specific viral respiratory pathogen, e.g. studies that only included patients with SARS-CoV-2 or influenza. We were therefore unable to present any pathogen specific results regarding prognostic factors for poor outcome. We do however agree with the reviewer that the included studies on LRTI patients are likely to include patients with a range of respiratory pathogens, both viral and bacterial. As such, it was not our intention to state that we could not include any studies on viral respiratory pathogens.

Changes: We have rephrased this limitation in order to better reflect our intended statement: 'Third, we excluded most studies on patients with a specific respiratory pathogen (e.g. SARS-CoV-2 or influenza) since these included patients with both upper and lower respiratory tract infections and did not report stratified results for LRTI. As such, we were unable to present any respiratory pathogen-specific results regarding prognostic factors for poor outcome in patients presenting to primary care with LRTI. However, since GPs are typically unaware of the specific pathogen in LRTI patients, such results would have limited applicability to current clinical practice.'

Reviewer #2

"The systematic review by Rijk et al. covers studies of prognostic factors and prediction models in lower respiratory tract infections in primary care. It summaries the studies descriptively by reporting on risk of bias, significant predictors, and performance of predictive models. The authors conclude that the papers reviewed are largely unreliable, but have some common important predictors amongst them.

Thank you to the editor for the opportunity to review this paper. It is well written and easy to follow, and is an important review that synthesizes the prognostic information in LTRIs. However, I have some questions surrounding the study design in relation to the aims that should be clarified in a revision."

Major Comments:

Comment 1. "The authors state that the aim is to "synthesize existing PFs and PMs for hospitalisation and mortality within 90 days in adult patients presenting to primary care with LRTI", but it is not clear for what purpose. The intro makes the point that it is important to predict whether a LRTI is viral or CAP, but the results include studies that only assess risk factors in CAP patients that have already been diagnosed. The intro also says that current indices (PSI & CURB-65) are no good for primary care, but the results include studies that investigated CRB-65 (It is not clarified if this predictive model also uses radiological/lab measures). A clearer aim on why the specific papers were reviewed should be considered."

Response: Upon reflection and prompted by this comment, we acknowledge that the initial Introduction section of our manuscript was strongly focused on LRTI aetiology (i.e. viral LRTI versus community-acquired pneumonia), whereas this was merely intended as an example of factors associated with poor outcome, such as hospitalization or mortality. Flagging high-risk patients may lead to an increased level of care or monitoring, thereby potentially enabling the mitigation of these outcomes. The aim of this systematic review is therefore to synthesize existing knowledge on prognostic factors and prediction models for hospitalisation and mortality in primary care LRTI

patients with a broad range of underlying conditions and causative respiratory pathogens. Such synthesis provides insight on the presence of (1) existing prognostic factors or prediction models suitable for use in clinical practice and (2) relevant prognostic factors to include when developing a new or updating an existing model.

Changes: As discussed above (response to comment #2 of reviewer #1), we have substantially revised the Introduction section to better reflect the aim and relevance of this systematic review.

Comment 2. "In line with the above it is not clear what the aim of reviewing both new PMs and the evaluation of standard PMs was. [...] If the aim was to synthesise the prognostic factors, the different models would help with that to understand what factors are included in and are important in each model. Conversely, the 5 studies evaluating the CRB-65 do not offer novel predictive information, but rather the synthesis of these studies would infer prognostic performance of that model and provide evidence of whether it should be used in primary care. Some clarification of the aims and purpose of the study within the intro would help address these points."

Response: As elaborated upon above (response to comment #1 of reviewer #2) the aim of our systematic review was to synthesize evidence on (1) existing prognostic factors or prediction models that might be suitable for use in clinical practice (see explanation on the potential clinical utility above) and (2) relevant prognostic factors that might be used when developing a new or updating an existing model developed prediction models. Hence, we have included various prognostic study designs in our synthesis: prognostic factor studies, prediction model development studies, and external validation studies.

Changes: Following the reviewer's comment we changed the Introduction section including the aims of the review, specified the eligible study designs in the Methods section and elaborated upon the reasons for including various prognostic study designs in the Discussion section (paragraph Strengths and limitations): 'By including various prognostic study designs, our synthesis provides a comprehensive overview of all prognostic evidence on primary care LRTI patients available. Studies assessing the prognostic value of individual factors, developing a new or validating an existing model all provide insight on important PFs that could be used in future prognostic research, whereas studies validating a new or previously developed model such as CRB-65 provide insight on the suitability for its use in clinical practice.'

Comment 3. "Page 11, line 44 says 9 different models were identified – but it appears that only actually 4 were identified, in addition to the CRB-65 that was evaluated in 5 different papers."

Response: We identified nine different prediction models, including some variants of existing models: CRB, CRB-65, CURB, CURB-65, PSI, a model developed by Bont et al., RISSC85, CORB-75, and a model developed by Moore et al. We do however acknowledge that the inclusion of some very similar but not identical prediction models might confuse readers.

Changes: We have explicitly stated the identified prediction models in the Results section: 'In total, nine different PMs could be included in the synthesis: CRB, CRB-65, CURB, CURB-65, PSI, a model developed by Bont et al., RISSC85, CORB-75, and a model developed by Moore et al.'

Minor Comments:

Comment 4. "CRB-65 should be defined in the abstract as a previously defined/standard model."

Response and changes: For clarity, we have defined CRB-65 as a previously developed prediction model in the revised abstract: '[...] whereas the previously hospital-derived CRB-65 was externally validated in primary care in five studies.'

Comment 5. "P11 Line 54 – Reported hospitalisation rates between 0.5 and 86.9% appears a big range for primary care. Should be clarified that this includes CAP only studies or separate CAP and LTRI from each other."

Response and changes: We agree that the large between-study differences in hospitalisation rates warrant further clarification. These large differences appear to be the result of varying study populations regarding LRTI diagnosis across studies, i.e. ranging from all LRTI patients to those

diagnosed with CAP. We therefore highlighted the study population with regards to diagnosis when providing hospitalisation rates in the Results section:

- For prognostic factor studies: 'The study population in terms of [...] the hospitalisation rate (from 0.7% in LRTI patients to 76.5% CAP patients) varied substantially across studies.'
- For prediction model studies: 'Hospitalisation rates varied between 0.5% in LRTI patients and 86.9% in CAP patients.'

Comment 6. "P11 Line 55-60 – The outcomes listed for each study does not add up to the overall number of PM studies (9)"

Response: We apologize for this confusion. In our synthesis of prediction model studies we have included seven different studies, which applied the following outcome definitions: mortality in three studies, hospitalisation in one study, a composite of both in one study and a broader outcome in two studies (i.e. re-consultation or hospitalisation and late-onset pneumonia, hospitalisation, or mortality). Although these seven studies provide evidence on nine different prediction models, the total number of outcomes applied in these studies is seven.

Comment 7. "P13 first paragraph – might be useful to show how many studies found the significant association for each predictor compared to how many studies investigated that predictor. For example: "multiple multivariable analyses were sex (five / five studies), current smoking (four /three studies, etc.)"

Response and changes: We thank the reviewer and, as suggested, we specified the number of studies that reported a significant association for each predictor in the Results section.

Comment 8. "Figure 2 is difficult to read. Perhaps placing the CAP and LTRI models side by side might be clearer, or summarising the predictor variables into a single row each, showing different colour points for different studies/categories."

Response and changes: We acknowledge that the initial version of Figure 2 was difficult to read. Following the reviewers' suggestion, we have now presented the prognostic factors of LRTI and CAP in a single row, indicating either LRTI or CAP patients based on coloured points.

Reviewer #3

"I have thoroughly reviewed your manuscript concerning prognostic factors for hospitalization and mortality in individuals with lower respiratory tract infections. I commend your efforts in addressing this critical topic. However, I would like to provide some feedback and suggestions to enhance the comprehensiveness and reliability of your findings."

Comment 1. "Literature Search and Inclusion of Non-English Publications and Grey Literature: It is crucial to understand whether the literature search conducted for this study was exhaustive. Did it include publications in languages other than English? Including non-English studies can provide a more comprehensive view, especially considering the global impact of lower respiratory tract infections."

Response: We excluded non-English articles (Method section, paragraph Searches and study selection). However, despite retrieval of four articles in languages other than English, none of the studies has been excluded based on language alone; the designs of these studies also did not match our review question. We therefore believe that our review provides a comprehensive overview of existing global literature in this field.

Comment 2. "Additionally, what grey literature was included in the study? This is particularly important in this field where health disparities, including those related to gender, age, socioeconomic status, living conditions, and prevalence of smoking, play a significant role. A broader scope in literature inclusion ensures that these disparities are well represented."

Response: We agree that grey literature can potentially be a valuable source of information when conducting a systematic review. Despite, we did not include grey literature sources in our search

strategy in this specific review. Since we identified only one additional record by citation and reference checking of included articles, we are however confident that our review provides a comprehensive overview of the global literature in this field. Also, it is highly unlikely that any additional grey literature, should this have been detected beyond our current search strategy, would have substantially influenced our findings given the (very) low certainty of evidence derived from the main electronic research databases for medicine and healthcare.

Comment 3. “Adjustment of Prognostic Factors Through Multivariate Analysis: It’s essential to emphasize whether these prognostic factors have been adjusted via multivariate analyses in the various studies reviewed. This adjustment is crucial to isolate the effect of each factor and understand its independent contribution to the risk of hospitalization and mortality.”

Response: In our presentation of the results of prognostic factor studies, we have provided effect estimates of both univariable (i.e. unadjusted) and multivariable (i.e. adjusted) analyses in separate supplementary figures (Figure S4 and S5, respectively). Figure 2 shows the most promising prognostic factors based on multivariable analysis as indicated in the upper part of the figure, although it is important to emphasize that the various studies did not use a standard set of covariates for this analysis.

Comment 4. “Baseline Risk Assessment: The manuscript should consider the baseline risk of hospitalization and death in patients with these infections. How is this baseline risk difference valued in your presentation? Presenting both relative and absolute estimates of the number of events with and without the risk factor, modelling different baseline risks (e.g., risk of death in children vs. elderly), can provide a clearer understanding of the real impact of these factors.”

Response: We agree with the reviewer that absolute risk estimates – in addition to relative estimates – are relevant when interpreting the impact of prognostic factors. As stated in our Results section (‘study characteristics’), Tables (Table 1 and 2), and Discussion section, the included studies were highly heterogeneous with regard to study population. As a result, baseline risk of hospitalisation and mortality varied substantially across studies (as presented in Table 1 and 2). In addition, most studies did not report absolute risks of outcome events stratified according to individual prognostic factors. Therefore, and because of the high between-study heterogeneity, we decided to present the baseline risks of hospitalisation and mortality (Table 1 and 2) and relative risk estimates of prognostic factors (Figure 2, S4, and S5) separately.

Comment 5. “Application of GRADE Methodology: The application of the GRADE methodology for each prognostic factor and outcome is recommended. This approach will help define the level of certainty we have for each effect estimate. This methodology is crucial for advancing towards prognostic scores and being able to estimate adjusted values and confidence in the effects for each estimation.”

Response and changes: We agree with the reviewer that the GRADE methodology is very useful to summarize the quality of evidence for and confidence in the individual effect estimates of systematic reviews. Prompted by this comment, we have applied the GRADE framework to the synthesis of promising prognostic factors (Table S6, supplementary material). The GRADE methodology and related findings have now been added to the Methods and Results sections, Figure 2 and Table S6, showing that the level of certainty of the evidence on promising prognostic factors is low to very low.

“Your research contributes significantly to understanding the dynamics of lower respiratory tract infections. These suggestions are intended to strengthen the robustness of your conclusions and enhance the paper’s utility for clinicians and policymakers.”

Response: We thank the reviewer for this positive remark and constructive comments which enabled us to further improve our manuscript and enhance the paper’s utility for clinicians and policymakers.

VERSION 2 – REVIEW

REVIEWER	Davido, Benjamin APHP, Maladies Infectieuses, Hôpital Universitaire Raymond-Poincaré
REVIEW RETURNED	12-Dec-2023

GENERAL COMMENTS	My reviewing included numerous suggestions that were all taken into account for clarification, and were approved by the authors. Congratulations for this work, and all the overhaul (that including 3 reviewers).
--

REVIEWER	Tortosa, Fernando Pan American Health Organization, EIH
REVIEW RETURNED	20-Dec-2023

GENERAL COMMENTS	I would like to begin by extending my sincere congratulations to the authors for conducting a well-executed review and for employing the GRADE approach in your analysis. The methodological rigor displayed in this article is commendable, and it's evident that considerable effort and expertise have been invested in this work. The use of GRADE, a renowned and systematic technique for evaluating the quality of evidence, particularly strengthens your study and sets a high standard for research in this field. In light of this, I believe there is an area that could benefit from further refinement. Specifically, the application of GRADE could be expanded not only to assess prognostic factors but also to evaluate modeled evidence, as outlined in the GRADE Series 30 reference. Enhancing the presentation of the various prognostic factors, by including both relative and absolute estimates, as well as the different baseline risks used for their calculation (presumably modeled in hospitalized populations), would add significant value and clarity to your findings. Moreover, applying a similar detailed approach to the modeled evidence would further solidify the robustness of your conclusions and provide a more comprehensive understanding for readers.
--

VERSION 2 – AUTHOR RESPONSE

Reviewer #1:

“My reviewing included numerous suggestions that were all taken into account for clarification, and were approved by the authors. Congratulations for this work, and all the overhaul (that included three reviewers). Regards.”

Response: We thank the reviewer for this positive comment and the time and effort spent on reviewing our manuscript.

Reviewer #3:

Comment 1: "I would like to begin by extending my sincere congratulations to the authors for conducting a well-executed review and for employing the GRADE approach in your analysis. The methodological rigor displayed in this article is commendable, and it's evident that considerable effort and expertise have been invested in this work. The use of GRADE, a renowned and systematic technique for evaluating the quality of evidence, particularly strengthens your study and sets a high standard for research in this field."

Response: We thank the reviewer for this very positive feedback.

Comment 2: "In light of this, I believe there is an area that could benefit from further refinement. Specifically, the application of GRADE could be expanded not only to assess prognostic factors but also to evaluate modelled evidence, as outlined in the GRADE Series 30 reference."

Response: We thank the reviewer for this suggestion.

Changes: As suggested, we have now applied the suggested GRADE guideline to rate the quality of evidence of included prediction models. This has now been added to the Methods and Results section and the results have been summarized in a new Table in the Supplementary data (Table S9).

Comment 3: "Enhancing the presentation of the various prognostic factors, by including both relative and absolute estimates, as well as the different baseline risks used for their calculation (presumably modelled in hospitalized populations), would add significant value and clarity to your findings."

Response: We agree with the reviewer that providing both relative and absolute risk estimates according to the various prognostic factors are instrumental for interpretation of the findings.

The baseline risks of the outcome (absolute estimates) in each prognostic factor study are presented in Table 1. The relative effect estimates of all included prognostic factors are presented in Figure 2, Figure S4, and Figure S6. Absolute risks according to the presence or absence of individual prognostic factors can only be calculated for univariable associations of categorical variables (i.e. not for continuous variables).

Changes: Where possible, based on the data from the primary studies, we have now provided the absolute risks according to the presence or absence of individual prognostic factors for univariable associations of categorical variables in a new Table in the Supplementary data (Table S5).

Comment 4: "Moreover, applying a similar detailed approach to the modelled evidence would further solidify the robustness of your conclusions and provide a more comprehensive understanding for readers."

Response: For all prediction model studies, baseline risks of the outcome have been provided in Table 2. We believe that presenting all absolute risks of all reviewed prediction models would be impractical and does not directly add to interpretation of prediction model performance. In addition, presenting absolute risks of prediction models is usually not possible since this requires access to individual patient data, which is hardly ever reported in prediction model studies. This is also the case for the included prediction model studies.

If reported in the primary studies, absolute risks as observed in the study cohort have been compared to the risks predicted by the respective prediction models, expressed as observed over expected (O/E) ratios. Measures of model calibration (e.g. O/E-ratios) provide a comprehensive overview of absolute and predicted risks over the full range of predictions and can be used to draw robust conclusions regarding the performance of the reviewed prediction models. As such, we feel that our current approach is most suited for our review objective.

VERSION 3 – REVIEW

REVIEWER	Tortosa, Fernando Pan American Health Organization, EIH
REVIEW RETURNED	07-Mar-2024
GENERAL COMMENTS	Thanks for the reviewers for the general and particular improvement of this manuscript.